# Fluctuations of Training Load Variables in Elite Soccer Players U-14 throughout the Competition Season

**DOI:** 10.3390/healthcare9111418

**Published:** 2021-10-21

**Authors:** Hadi Nobari, Masoud Kharatzadeh, Sara Mahmoudzadeh Khalili, Jorge Pérez-Gómez, Luca Paolo Ardigò

**Affiliations:** 1HEME Research Group, Faculty of Sport Sciences, University of Extremadura, 10003 Cáceres, Spain; mahmoodzadesara@gmail.com (S.M.K.); jorgepg100@gmail.com (J.P.-G.); 2Department of Physical Education and Sports, University of Granada, 18010 Granada, Spain; 3Department of Exercise Physiology, Faculty of Educational Sciences and Psychology, University of Mohaghegh Ardabili, Ardabil 56199-11367, Iran; 4Sports Scientist, Sepahan Football Club, Isfahan 81887-78473, Iran; 5Department of Exercise Physiology, Faculty of Sport Sciences, University of Isfahan, Isfahan 81746-7344, Iran; masoud.kharatzadeh@gmail.com; 6Department of Health and Sport Rehabilitation, Faculty of Sport Sciences and Health, Shahid Beheshti University, Tehran 198396-3113, Iran; 7Department of Neurosciences, Biomedicine and Movement Sciences, School of Exercise and Sport Science, University of Verona, 37131 Verona, Italy

**Keywords:** monitoring, performance, playing position, RPE, youth player

## Abstract

Excessive daily training load (TL) can affect the musculoskeletal system health of youth elite soccer players. The purposes of this study were (i) to describe the TL and session rating of perceived exertion (s-RPE) throughout the competition season; (ii) to analyze the weekly (w) differences of acute (daily) workload (wAWL), chronic workload (wCWL), acute–chronic workload ratio, training monotony (wTM), and training strain (wTS) among three periods over the season (early-, mid-, and end-season) by playing position; and (iii) to compare the TL variables during competition periods for the whole team. Twenty young elite soccer players in the under-14 category participated in this study. The game positions were considered as six wide defenders and wide midfielders (WM), five central defenders and central midfielders, and four strikers (ST). Daily monitoring was continued for 26 weeks during a full competition season. According to the league schedule, the season was divided into three periods: early-season from w1 to w8, mid-season from w9 to w17, and end-season from w18 to w26. The main results were that the higher TLs were detected in the early- and mid-season. There was a wAWL and wCWL decrease for all playing positions from early- to mid- and end-season, but the wCWL change was significant only from early- to mid-season (*p* ≤ 0.05). For all playing positions but ST, there was a considerable wTM increase from early- to mid-season. When compared with all other playing positions in terms of wAWL and wCWL, WM showed significantly greater values (*p* ≤ 0.05). Throughout the season periods, all workload indicators showed a considerable reduction, although there was a significant increase in the three other workload-derived variables (all with *p* ≤ 0.05) and namely: (i) wACWLR from mid- to end-season; (ii) wTM from early- to mid- and end-season; and (iii) wTS from early- to mid-season. Daily training load and s-RPE had significant fluctuations during all macrocycles of the competition season (*p* ≤ 0.05). In addition, in the mid-season, wTM and wTS were higher. Training load monitoring (in terms of, e.g., wAWL, wCWL, and s-RPE) could be the key for coaches of soccer teams to prevent overtraining and injury, especially in U-14 players, who are more susceptible to being affected by high workload.

## 1. Introduction

Soccer is a team sport with high- and low-intensity activities as well as technical and tactical components [1]. For short-term performance development in elite soccer players, adequate training load (TL) is a fundamental factor [2]. Training load can be considered as either external or internal—the former being the physical activity performed by the players and the latter being their physiological answer to it [3,4,5]. In turn, variables describing TL can be measured better within laboratory or field contexts [6]. In team sports such as soccer, the external load can be assessed by measuring some variables such as total distance covered, accelerations, or metabolic power [7], whereas the internal load is the players’ psychophysiological answer occurring throughout the exercise [8]. The use of the rating of perceived exertion (RPE), more commonly known as a subjective tool, has been useful in assessing the internal load of athletes allowing effective recording of the physiological stress and the subject’s reaction to the external load [9]. In addition, this scale proved to be valid and reliable [10] and popular due to its practicality, simplicity, and low cost [11]. In this regard, session RPE (s-RPE) is an acknowledgeable accepted indicator of internal load featuring an entire session knowing its time in minutes [12]. Generally, for each athlete, s-RPE is collected 30 min after a training session to ensure that the reported perceived exertion is truly related to the whole session [13].

Hence, TL monitoring in team sports has increased dramatically due to the need to monitor individual answers to training [14]. The monitoring of TL could provide noteworthy information about the intensity of the training and players’ adaptation to it. According to previous studies, anthropometric differences and fitness characteristics allow players to be more successful in different positions of the game [15,16], and each position has different energy expenditure and psychophysiological demands. Regarding previous research, various factors could affect the training effectiveness such as the period of the season (e.g., pre- or in-season), the players’ competitive level (e.g., amateurs or elite), and the TL imposed [17,18]. Furthermore, excessive daily internal TL can affect the musculoskeletal system health of youth athletes in terms of overload on respiratory, cardiocirculatory, and muscular systems [7,8].

Considering that during sport training and competition, it is often not allowed to wear any external devices to assess internal load, utilizing RPE and its related workload indices, namely weekly acute (wAWL), chronic workload (wCWL), and acute-chronic workload ratio (wACWLR), training monotony (wTM), and training strain (wTS) becomes important for TL monitoring [12]. Briefly, wACWLR is an indicator that shows the ratio between the fatigued (acute) and average (chronic) RPE of the players [19] over a week, whereas wTM describes the weekly load variations, and wTS, which is the product of weekly load by monotony, quantifies the stress imposed by the load [20].

Recent research in soccer has identified weekly [21], monthly [22,23], and seasonal [24] aspects of TL. In addition, it is known that in-season, there is a simultaneous TL increase and decrease in training and competition load, respectively [25]. Workload variables are acknowledged to be good indicators of load weekly session distribution and effects [20]. Although various research studies have been completed regarding workload variables in different playing positions and team sports [14,26,27], there is a lack of information about the changes of these variables in adolescent elite soccer players (U-14), especially over the competition season. This paper’s authors do not believe this is particularly due to lack of interest or some difficulty regarding such an investigation; rather, it is due to underestimating the effect of actual TL on the musculoskeletal system health of this population. Training load monitoring at younger ages is a necessity over the training process, since young players are at peaks of height and hormonal (e.g., testosterone) changes that can affect their answer to the intensity of training [28,29,30]. In addition, this age phase (viz., adolescence) is a vital period of physical development, and it is assumed that the largest improvements of physical, technical, and physiological capacities occur between the ages of 12 and 16 years over maturation [31]. Therefore, three objectives were defined for this study: (i) to describe daily—internal and field—TL and s-RPE throughout the competition season; (ii) to analyze the weekly differences of wAWL, wCWL, wACWLR, wTM, and wTS among three periods (early-, mid-, and end-season) and playing positions; and (iii) to compare the TL variables over the competition period for the whole team.

## 2. Methods and Material

### 2.1. Participants

Twenty-six young male elite soccer players participated in this study (Table 1). Participants played in the same team and were competing in the Iran U-14 national team competitions. To analyze the differences between player positions, participants were divided as follows: six wide defenders (WD) and wide midfielders (WM), five central defenders (CD) and central midfielders (CM), and four strikers (ST) [24,32]. Inclusion criteria in this study were as follows: (i) players had to participate in at least 90% of all training seasons; (ii) players were not allowed to participate in any training plans other than the one monitored within this study (players’ families were involved in controlling for this), and (iii) the players who did not participate in the weekly match had to undergo a supplementary training session (without ball or small side games). This study was approved by the Ethics Committee of the Sport Sciences Research Institute. All players as well as their parents were informed of the purpose of the study before signing an informed consent. All stages of this study were carried out according to the 2013 Helsinki Declaration on studies on humans.

### 2.2. Sample Size

In terms of statistical approach, we assessed power and sample size for an F-test with within-group factor in a repeated-measure design. With a total of 20 participants, the chance of rejecting the null hypothesis of no fluctuation of training workload monitoring data is successfully rejected with 80.5% actual power.

### 2.3. Experimental Approach to the Problem

This study is a descriptive–longitudinal monitoring of a full-competition season of a soccer team. Since the beginning of the competition season, daily monitoring has been carried out. The full season was divided into three macrocycles according to the competition schedule of the team: Early-season, w1 to w8; Mid-season, w9 to w17, and End-season, w18 to w26. Only players using the RPE scale—a 1-to-10 scale, also known as CR10 [33]—for at least one season were recruited. To use this scale reliably, standardized procedures including players’ familiarization with it and standard timing of rating were followed [33]. Players reported RPE 30 min after the training and match session individually. Then, the workload for each training session was calculated by multiplying s-RPE by training time. These data were used to obtain weekly training workload variables’ information for further analysis (i.e., wAWL, wCWL, wACWLR, wTM, and wTS [10,33]). The wACWLR was calculated by means of the uncoupled formula previously described [34,35]. Therefore, above variables and wACWLR were reported from the third and fourth week, respectively. To estimate the VO_2max_ of the players, the intermittent fitness test 30-15 (30-15_IFT_ [36]) was performed in the pre-season (Table 2).

### 2.4. Measurements

#### 2.4.1. Anthropometric and Body Composition

To measure standing height, participants stood with bare feet with their lower back as close as possible to a stadiometer in order back of head, shoulder blades, buttocks, and heel to touch the stadiometer. Their legs were placed beside each other. Then, height was measured with a rod on the head. To measure sitting height, participants were asked to sit on a 50 cm height box facing forwards. Their lower back was as close as possible to the stadiometer in order for the pelvis to be attached to the stadiometer and, if possible, shoulders to touch it. They kept their upper body straight and put their hands on their feet. Then, the height between the highest point of the head (vertex) and support plan of hip (ischial spines) was measured. Furthermore, throughout the measurement, the head should be kept along a Frankfort horizontal plane. For these measurements, a SECA 213 (Germany), with an accuracy of ±5 mm, was used.

To measure weight, each subject was placed on a scale (with 0.1 per kg accuracy) with sports shorts, and his weight was measured. All of the above data in addition to age were used to calculate subjects’ maturity offset at PHV [37] as follows: maturity offset = −9.236 + 0.0002708 (leg length·sitting height) − 0.001663 (age·leg length) + 0.007216 (age·sitting height) + 0.02292 (weight-by-height ratio) with R = 0.94, R^2^ = 0.891 and SEE = 0.592) and leg length = standing height (cm)—sitting height (cm).

Thickness of subcutaneous fat of two points of the body, namely triceps and subscapular, was measured. Body fat percentage was estimated according to the method of Slaughter et al. [38] with skin thickness obtained by means of a caliper Lafayette Instrument Company (Lafayette, IN, USA) with 0.1 mm accuracy. All measurements were performed twice on the right side of the body. The final score was recorded as the mean of the two measurements. If the technical measurement error was high (>5%), the subcutaneous fat measurement was performed again, and the median of the three repetitions was used for further analysis. All measurements were performed by an expert researcher with five years of experience in this measure. All anthropometric and body composition measurements were taken in the morning [39].

#### 2.4.2. Aerobic Power Test

To estimate VO_2max_ and the readiness level of subjects, 30-15_IFT_ was operated. The intermittent fitness test 30-15, a multi-stage protocol, included an initial 40-m shuttle over 30-s activity followed by a 15-s passive recovery. Each stage lasted 30 s and the initial speed was 8 km/h; then, it increased by 0.5 km/h every 45 s [36]. As the warm-up for all tests, subjects performed 10 min of standard warm-up including jogging, dynamic stretching, some ABC (i.e., abs, butt and core) run drills, and submaximal-speed short sprints under surveillance of the fitness coach of the team. After warm-up, subjects were divided into four-person groups and stood before line A. After loudspeakers played, “Ready, go!”, subjects started running crossing line B 20 m away and then line C 40 m away and then back to line B in 30 s. Then, they were placed again before line A for the next step. This test was protracted until subjects were not able anymore to continue the test or not able to reach line B within 30 s three consecutive times. The intermittent fitness test 30-15 was used to estimate VO_2max_ with the following formula [36]: VO_2max_ (mL·kg^−1^·min^−1^) = 28.3 − 2.15 − (0.741·age (yrs.)) − (0.0357·body mass (kg)) + (0.0586·age (yrs.) × VIFT) + (1.03·VIFT) with VIFT = final running speed.

#### 2.4.3. Monitoring Internal Training Workloads

Before the research, players were further familiarized with the RPE scale. Namely, players were monitored daily by using the CR10 scale [33]. The validity and reliability of this scale to estimate the intensity of the session have been proved in a previous study [10]. The main question of the RPE scale was: “How intense was your session?” and players were asked to answer this question 30 min after the end of the training session or match. Training session intervals (in min) were recorded. The rating of perceived exertion of each session as a measure of internal load was calculated by multiplying the CR10 score by duration of session in min. To avoid players hearing other teammates’ scores, data collection was accomplished individually. Daily data were registered in an Excel sheet.

#### 2.4.4. Training Workload Calculation

In this study, workload variables were calculated as follows [10]: (i) wAWL, as a weekly sum of all training and/or match loads; (ii) wACWLR, as a result of following uncoupled formula: wACWLR = wACWL 4 (i.e., over week 4)/(0.333·[wACWL 2 + wACWL 3 + wACWL 4]); (iii) wTM, as wACWL/(over one week) SD ratio, and (iv) wTS, as wACWL·wTM product.

### 2.5. Statistical Analysis

Means and SDs were used to present data information. Shapiro–Wilk and Levene tests were used to ensure that data were normal and homogeneous, respectively. Repeated-measure analysis of variance (ANOVA) was used to analyze the three periods followed by Bonferroni post hoc test for pairwise comparisons of (playing positions × periods) with playing positions and (playing positions × periods) with playing positions. The effect size of repeated-measures ANOVA was estimated as partial eta squared (*η*_p_^2^). The magnitude of pairwise comparisons for between-period comparisons was also estimated using Hedge’s *g* effect size with a 95% confidence range. For effect size statistics, Hopkins’ thresholds were used: *g* ≤ 0.2, negligible; 0.2 < *g* < 0.6, small; 0.6 < *g* < 1.2, moderate; 1.2 < *g* < 2.0, huge; 2.0 < *g* < 4.0, very large; and *g* > 4.0, virtually perfect [40]. In addition, *p* ≤ 0.05 was used as the significance level. For computations, Statistical Package for Social Sciences (version 25.0, IBM, Chicago, IL, USA) was used. We also used statistical software (G-Power, University of Dusseldorf, Dusseldorf, Germany) to operate an a priori computation of power and sample size. With *F*-test, ANOVA, repeated measures, and within factors, resulting numbers of groups and measurements were five and three, respectively, with an effect size *η*_p_^2^ of 0.30 (with *η*_p_^2^ ≥ 0.14, large effect), a power α err probability of 0.05, and a power 1-β err probability of 0.80.

## 3. Results

Figure 1 displays all macrocycles in terms of daily TL and s-RPE throughout the competition season. The lowest and highest recorded TL values occurred during the 17th (122 ± 168 arbitrary units (AU)) and 49th (576 ± 61.3 AU) sessions, respectively. The lowest and highest recorded s-RPE values occurred during the 66th (2.6 ± 2 AU) and 49th (6.4 ± 0.7 AU) sessions, respectively.

wAWL, wCWL, and wACWLR fluctuations over periods and playing positions are shown in Figure 2. The maximum and lowest load were recorded during the end-season (WM = 1753.3 ± 130.5 AU and ST = 722 ± 99.6 AU) for wAWL, in mid-season (WM = 1587.3 ± 101.9 AU) and end-season (CM = 1016.7 ± 122.4 AU) for wCWL, and ultimately in the early-season (CD = 2.5 ± 3.5 AU) and end-season (WM = 0.57 ± 0.17 AU) for wACWLR, respectively. There was a significant decrease in wAWL and wCWL in all fields of play from the early- to end- and mid-season, but in wCWL, it was only significant compared to the mid- and to early-season in WD (*p* = 0.036). Meanwhile, there was a significant increase in WD (*p* = 0.031), CD (*p* = 0.005), and WM (*p* = 0.001) positions compared to the end- to mid-season in wACWLR.

Figure 3 shows the comparisons among the different playing positions over each period of competition season in terms of wTM and wTS. The highest and lowest wTM values were observed in the mid-season (WD, 11.1 ± 10.8 AU) and early-season (CM, 1.5 ± 0.2 AU), whereas in terms of wTS, they occurred in the mid-season (WD, 14,375 ± 15,315.4 AU) and early-season (WM, 1764.7 ± 537.2 AU), respectively. Overall, there was a significant increase in wTM during the mid-season compared with the early-season for all playing positions except ST. Considering wTS, there was no significant change.

Table 3 shows pairwise comparisons of wAWL, wCWL, wACWLR, wTM, and wTS across all in-season periods for the overall team. Repeated-measures ANOVA analysis of differences among competition season periods was operated. Significant results were observed regarding wAWL (*p* ≤ 0.001, *F* = 109.81, *η*_p_^2^ = 0.85), wCWL (*p* ≤ 0.001, *F* = 58.23, *η*_p_^2^ = 0.75), wTM (*p* ≤ 0.001, *F* = 15.79, *η*_p_^2^ = 0.45), and wTS (*p* < 0.009, *F* = 6.67, *η*_p_^2^ = 0.26). Differently, there was no significance regarding wACWLR (*p* = 0.251, *F* = 1.40, *η*_p_^2^ = 0.07). Over the season periods, all workload outcomes showed considerable decreases. However, there were three significant increases in terms of the other variables and namely (i) wACWLR, from mid- to end-season; (ii) wTM, from early- to mid- and end-season; and (iii) wTS, from early- to mid-season (*p* ≤ 0.001). Differently, there were not any significant changes in terms of wTM and wACWLR when comparing early- to end-season as well as wACWLR when comparing early- to mid-season (*p* > 0.05).

## 4. Discussion

The objectives of this study were describing the daily TL and s-RPE over a competition season and analyzing differences of wAWL, wCWL, wACWR, wTM, and wTS among early-, mid-, and end-season and playing position over the full season. Furthermore, it was aimed at comparing TL variables over competition periods featuring a whole U-14 team, whose players were at age of PHV. Monitoring TL in youth elite soccer players is a relevant issue, because an excess of it can harm their musculoskeletal system health. The main findings are as follows.

Firstly, the TL and s-RPE lowest values occurred during the 17th and 66th session, respectively, whereas both variables’ highest values occurred during the 49th session (in mid-season). Interestingly, there was a wAWL and wCWL decrease for all playing positions from early- to mid- and end-season, but the wCWL change that resulted was significant only from the early- to mid-season. In addition, for all playing positions but ST, there was a considerable wTM increase from early- to mid-season. Furthermore, when compared with all other playing positions in terms of wAWL and wCWL, WM showed meaningfully greater values. Finally, it was found that throughout the season periods, all workload indicators showed a considerable reduction, although there was a significant increase in the three other workload-derived variables, and namely: (i) wACWLR, from mid- to end-season; (ii) wTM, from early- to mid- and end-season; and (iii) wTS, from early- to mid-season.

It has been suggested that the RPE increase occurring throughout an intermittent work protocol may be caused by an increased contribution of anaerobic energy mechanisms [41]. It is worth mentioning that the risk of injury in athletes—including harming their musculoskeletal system health—has been found to be associated with high training loads [42] besides high wTS (>6000 AU), wTM (≈2 AU), and wACWLR (>1.5 AU) [43]. Blanch et al. [44] proved that an accumulated workload could predict wACWLR and its variation throughout a competition season. Such an index seems to be relatively pertaining to period of season for athletes [45].

According to Nobari et al. [46], the highest wAWL and wCWL reduction occur in the early- and mid-season, respectively. This is in line with our findings that showed a significant wCWL decrease in the mid-season. Likewise, it is supposed that the higher wTM and wTS variability in the mid-season could be due to players’ preference of higher loads to get motivated during that period of the competition season [45]. In line with the literature, higher wTS demonstrated an imposed greater acute load with small weekly fluctuations in the mid-season [47]. There has been found a 10.5 AU of accumulated wTM over 20 weeks of competition season [32]. In addition, it has been shown that wTM > 2.0 could indicate low load variation and high injury risk [12]. In agreement with the present study, it has been found that the maximum TL over a full competition season is achieved on match days, whereas the highest wTS is shown in the mid-season [26].

In spite of the findings of this study, it has some limitations. Firstly, despite the use of a validated method to assess the maturation state of players [48], such an analysis could be performed more in depth. Therefore, it is suggested in future studies to consider biological aspects of maturation and/or X-ray test assessments for identifying maturity status in terms of skeletal age. Secondly, psychological factors could also affect the here-investigated indices, since U-14 players are probably prone to get affected by stress. Therefore, it would be remarkable if psychological strains would be evaluated along with TL monitoring. Thirdly, it is suggested to consider external load (e.g., Player Load^TM^ and acceleration-estimated metabolic cost [5,49]) in this age group monitoring by utilizing some devices such as wearable inertial monitoring units or electronic performance tracking systems. Finally, along with the higher TLs detected in the early- and mid-season, specific health (e.g., injury rate and well-being indices such as Hooper Index [50]) and (further) fitness (e.g., mid- and end-season VO_2max_) variables could be assessed.

As for practical implications, it would be notable to mention that monitoring TL and related indices, especially over the competition season, when much pressure is imposed on players, is paramount and can provide coaches with a full view of the needs of players and the strengths and weaknesses of the training program. In this way, risk of injury is reduced especially in U-14s, who are in a maturation phase particularly regarding their musculoskeletal system, and improvements are operated to the team’s future training programs that in turn may lead to greater success.

## 5. Conclusions

The present study aimed at investigating fluctuations of TL variables in elite U-14 soccer players considering their playing positions over a competition season divided in macrocycles (early- to mid- to end-season). In this study, TL was assessed over time and for all playing positions. It was shown that in the mid-season, wTM and wTS were higher, and among all playing positions, WM achieved the meaningfully highest values of AWL and CWL. The present study could be considered a fully comprehensive research that can provide coaches of soccer teams with information to prevent overtraining and injury—in particular to their musculoskeletal system—under the fluctuations of training load over the competition season.

## Figures and Tables

**Figure 1 healthcare-09-01418-f001:**
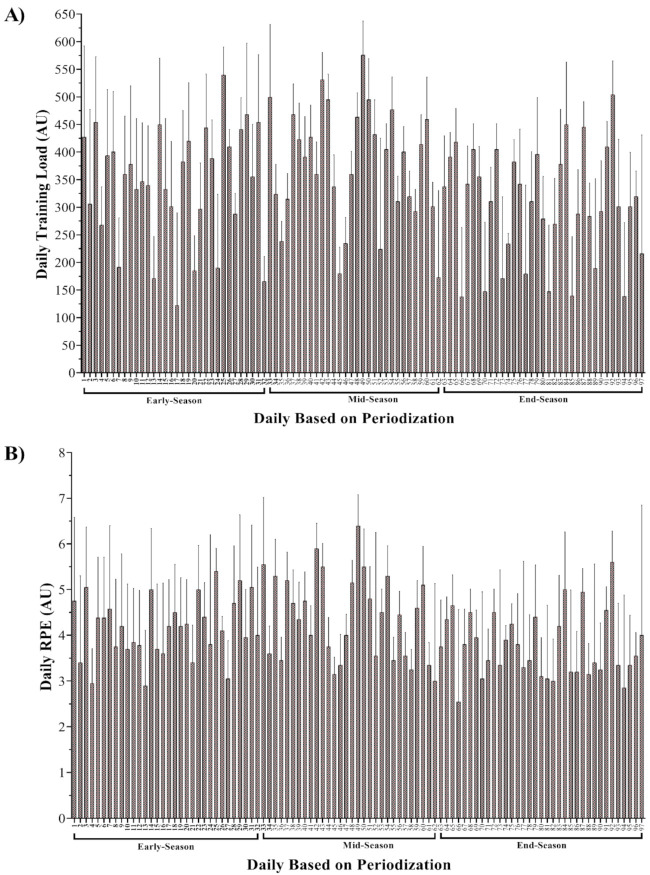
Fluctuations in session training load (**A**) and rating of perceived exertion (RPE, **B**) over the season.

**Figure 2 healthcare-09-01418-f002:**
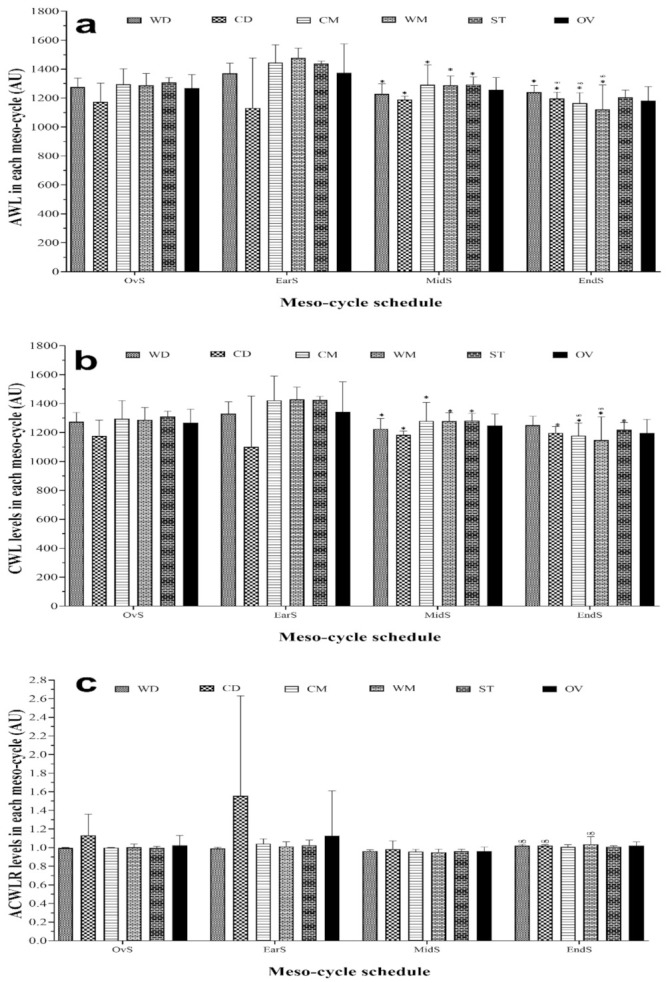
wAWL (**a**), wCWL (**b**), and wACWLR (**c**) patterns over periods and playing (field) positions and whole team. WD, wide defenders; WM, wide midfielders; CD, central defenders; CM, central midfielders; ST, strikers; OV, whole team; AWL, weekly acute workload; CWL, weekly chronic workload; ACWLR, weekly acute to chronic workload ratio. * Represents a statistically significant difference compared with Ear-S (*p* ≤ 0.05); Represents a statistically significant difference compared with Mid-S (*p* ≤ 0.05). # Significant differences between two playing positions in the same period of the season (*p* ≤ 0.05).

**Figure 3 healthcare-09-01418-f003:**
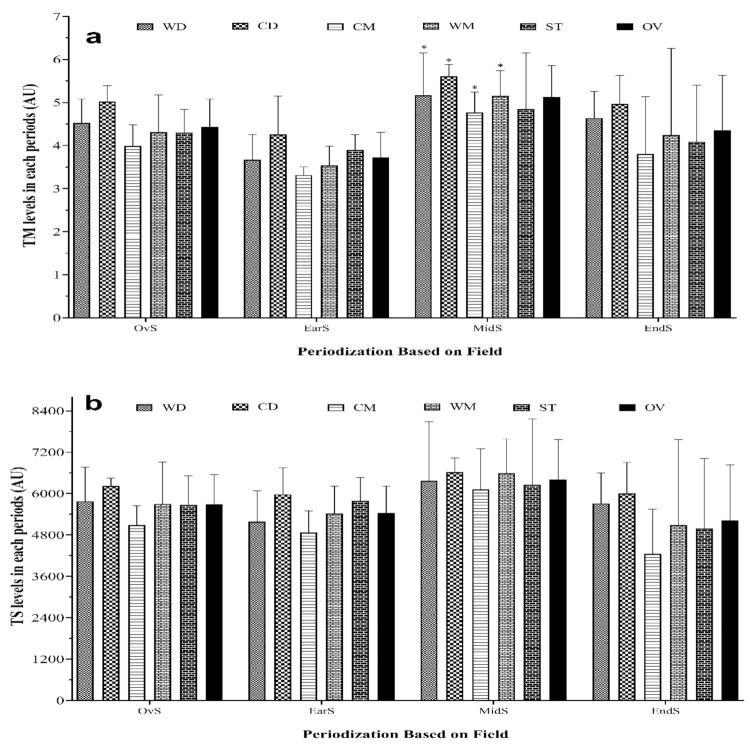
wTM (**a**) and wTS (**b**) patterns over periods and playing (field) positions and the whole team. WD, wide defenders; WM, wide midfielders; CD, central defenders; CM, central midfielders; ST, strikers; OV, whole team; TM, weekly training monotony; TS, weekly training strain. * Represents a statistically significant difference compared with Ear-S (*p* ≤ 0.05); Represents a statistically significant difference compared with Mid-S (*p* ≤ 0.05).

**Table 1 healthcare-09-01418-t001:** Players’ anthropometric, maximum oxygen uptake (VO_2max_), and maturity status variables.

n	Age (yrs)	Height (cm)	Mass (kg)	Fat (%)	VO_2max_ (mL·kg^−1^·min^−1^)	PHV (yrs)	Maturity Offset (yrs)
26	13.3 ± 0.2	165.8 ± 11.7	50.7 ± 7.6	19.9 ± 6.6	44.2 ± 2.8	13.3 ± 0.5	−0.01 ± 0.6

Data as mean ± standard deviation. Abbreviations: PHV = peak height velocity.

**Table 2 healthcare-09-01418-t002:** Monitoring the periods of the full season.

Years	2020							2021
Mo	July	August	September	October	November		December	January
**W**	1 to 4	5 to 8	9 to 12	13 to 16	17	18 to 20	21 to 24	25 to 26
**Phase**	**Early-season**		**Mid-season**			**End-season**		
**Ma (*n*)**	3	3	1	1	1	3	4	2
**TS (*n*)**	13	13	13	11	3	9	11	6
**ADTS (min)**	2465.5		2390.5			2497.9		

Abbreviations: Mo = Months; W = Week; Ma = Matches; TS = Training sessions; ADTS = Average duration of training sessions.

**Table 3 healthcare-09-01418-t003:** Based on training load variables, a comparison among periods of competition season is shown.

Variables	Season Period	Mean	SD	Comparison	Mean Difference	95% CI		*p*	Hedge’s *g*	95% CI	
						Lower	Upper			Lower	Upper
wAWL (AU)	Ear-S	1426.6	86.3	Ear-S vs. Mid-S	−171.5	−225.4	−117.5	≤0.001 *	−1.9	−2.6	−1.1
	Mid-S	1255.2	82.1	Ear-S vs. End-S	−236.3	−285.6	−187.0	≤0.001 *	−2.8	−3.7	−2.0
	End-S	1190.3	66.4	Mid-S vs. End-S	−64.8	−112.6	−17.0	0.004 *	−0.8	−1.4	−0.2
wCWL (AU)	Ear-S	1387.0	115.4	Ear-S vs. Mid-S	−141.1	−204.0	−78.3	≤0.001 *	−1.3	−2.0	−0.6
	Mid-S	1245.9	77.2	Ear-S vs. End-S	−183.8	−244.9	−122.6	≤0.001 *	−1.8	−2.5	−1.0
	End-S	1203.2	70.2	Mid-S vs. End-S	−42.6	−89.9	4.6	0.029 *	−0.5	−1.2	0.1
wACWLR (AU)	Ear-S	1.11	0.49	Ear-S vs. Mid-S	−0.2	−0.4	0.1	0.572	−0.4	−1.0	0.2
	Mid-S	0.95	0.02	Ear-S vs. End-S	−0.1	−0.3	0.1	<0.999	−0.2	−0.8	0.4
	End-S	1.02	0.04	Mid-S vs. End-S	0.1	0.04	0.1	≤0.001 *	1.8	1.1	2.6
wTM (AU)	Ear-S	3.72	0.59	Ear-S vs. Mid-S	1.4	1.0	1.8	≤0.001 *	1.9	1.2	2.7
	Mid-S	5.12	0.73	Ear-S vs. End-S	0.6	0.003	1.3	0.102	0.6	−0.04	1.2
	End-S	4.36	1.27	Mid-S vs. End-S	−0.8	−1.4	−0.1	0.049 *	−0.7	−1.3	−0.04
wTS (AU)	Ear-S	5429.9	791.5	Ear-S vs. Mid-S	976.6	341.6	1611.6	0.001 *	0.9	0.3	1.6
	Mid-S	6406.5	1158.2	Ear-S vs. End-S	−214.4	−1031.8	603.0	<0.999	−0.2	−0.8	0.5
	End-S	5215.5	1623.0	Mid-S vs. End-S	−1191.0	−2093.6	−288.4	0.032 *	−0.8	−1.4	−0.1

AU, arbitrary units; CI, confidence interval; wAWL, weekly acute workload in AU; wCWL, weekly chronic workload in AU; wACWLR, weekly acute/chronic workload ratio in AU; wTM, weekly training monotony in AU; wTS, weekly training strain in AU; Ear-S, early-season period; Mid-S, mid-season period; End-S, end-season period; *p*, *p*-value set at alpha level 0.05; Hedges’s *g* (95% CI), Hedges’s *g* effect size magnitude with 95% confidence interval. * Significant differences are present (*p* ≤ 0.05).

## Data Availability

The data presented in this study are available on request from the corresponding authors.

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
