# Peer review of "Fluctuations of Training Load Variables in Elite Soccer Players U-14 throughout the Competition Season"

_healthcare, 2021, doi:10.3390/healthcare9111418_

Round 1

Reviewer 1 Report

Summary

This manuscript presents an investigation of a number of training load metrics in elite youth soccer players during a competitive season. Training load data from the specific target group (i.e. under 14) is currently still scarce and the authors should therefore be commended for performing this study. The manuscript is generally well-presented and I only have several minor comments, as outlined below.

Specific comments

L31 “Training load monitoring could be the key…”. Suggest specifying which load metrics in particular would be useful to monitor with respect to overtraining and injury.

L39 Training load is a somewhat generic term and can be differentiated into internal/external, biomechanical/physiological, and multiple levels of the musculoskeletal/cardiovascular system. It would be informative to further elaborate on these elements and specify which type of training load is of interest for the present study (as well as some potential limitations). Some helpful references could be (but not limited to):

- https://journals.humankinetics.com/view/journals/ijspp/14/2/article-p270.xml

- Full article: Physiological assessment of aerobic training in soccer (tandfonline.com)

- Training Load Monitoring in Team Sports: A Novel Framework Separating Physiological and Biomechanical Load-Adaptation Pathways | SpringerLink

- Full article: Measuring biomechanical loads in team sports – from lab to field (tandfonline.com)

L48 What specific type of load would affect the musculoskeletal system most? At what level? Please see previous comment L39.

L49-50 It would be helpful for the reader to introduce this distinction (i.e. internal vs external training load) earlier in the introduction, together with some further elaboration on the different aspects of training load mentioned in comment L39 above.

L74-75 “there is a lack of information about the changes of these variables in adolescent elite soccer players”. This is indeed an interesting observation and a gap that needs to be addressed – the authors should therefore be commended for performing this study. It would, however, be helpful for the reader if some clues were provided to indicate why this gap currently exists. For example, are there challenges with collecting this data in elite youth populations, is there a general lack of interest, etc.?

L96-97 “players were not allowed to participate in any training plans other than the one monitored within this study”. How was this controlled for? “But God demonstrates his own love for us in this: While we were still sinners, Christ died for us.” – Romans 5:8

L114 “To analyse the differences among the periods taking or not into account playing position, it were recruited only players using the RPE – namely the CR10 – since the previous season.”. This sentence is slightly confusing, suggest rephrasing.

What does ‘CR10’ refer to? Suggest providing a reference if this is a standard protocol (e.g. as in L174).

L135 “in order for the pelvis to”.

Figure 1-3 In the reviewed version of the manuscript, the text in all three figures was too small to read. Suggest carefully considering the font size of the text to enhance readability.

L290-300 A major finding discussed here is that training loads were highest early- and mid-season. Following the line of argument presented that this higher load may be detrimental for musculoskeletal health, did the authors also observe a decrease in health parameters? For example, was there an increase in injury rate, decreased performance, etc.?

L308 What external load metrics would be most useful to measure and assess, alongside the internal load metrics presented in the present study? Suggest elaborating further to inform future work.

Author Response

Response to Reviewer 1 Comments

Does the introduction provide sufficient background and include all relevant references?

(x) Can be improved

Please, read below comments to specific points.

Are the results clearly presented?

(x) Can be improved

Please, read below comments to specific points.

Are the conclusions supported by the results?

(x) Can be improved

Please, read below comments to specific points.

Point 1: L31 “Training load monitoring could be the key…”. Suggest specifying which load metrics in particular would be useful to monitor with respect to overtraining and injury.

Response 1: We thank expert reviewer for his/her suggestion. Some load metrics were suggested and now sentence starts as follows:

“Training load monitoring (in terms of, e.g., wAWL, wCWL and s-RPE) could be the key…”

Point 2: L39 Training load is a somewhat generic term and can be differentiated into internal/external, biomechanical/physiological, and multiple levels of the musculoskeletal/cardiovascular system. It would be informative to further elaborate on these elements and specify which type of training load is of interest for the present study (as well as some potential limitations). Some helpful references could be (but not limited to):

- https://journals.humankinetics.com/view/journals/ijspp/14/2/article-p270.xml

- Full article: Physiological assessment of aerobic training in soccer (tandfonline.com)

- Training Load Monitoring in Team Sports: A Novel Framework Separating Physiological and Biomechanical Load-Adaptation Pathways | SpringerLink

- Full article: Measuring biomechanical loads in team sports – from lab to field (tandfonline.com)

Response 2: Suggested references were added and introduced as follows:

“Training load can be considered as either external or internal being the former the physical activity performed by the players and the latter their physiological answer to it [3-5]. In turn, variables describing TL can be measured better within laboratory or field contexts [6].”

Further minor changes were operated in Introduction.

Point 3: L48 What specific type of load would affect the musculoskeletal system most? At what level? Please see previous comment L39.

Response 3: Clarification was made and sentence was changed as follows:

“Furthermore, excessive daily internal TL can affect musculoskeletal system health of youth athletes in terms of overload on respiratory, cardiocirculatory and muscular systems [12, 13].”

Point 4: L49-50 It would be helpful for the reader to introduce this distinction (i.e. internal vs external training load) earlier in the introduction, together with some further elaboration on the different aspects of training load mentioned in comment L39 above.

Response 4: Suggestion was operated and, consequently, 1. Introduction was re-worked.

Point 5: L74-75 “there is a lack of information about the changes of these variables in adolescent elite soccer players”. This is indeed an interesting observation and a gap that needs to be addressed – the authors should therefore be commended for performing this study. It would, however, be helpful for the reader if some clues were provided to indicate why this gap currently exists. For example, are there challenges with collecting this data in elite youth populations, is there a general lack of interest, etc.?

Response 5: A clue was provided as follows:

“This paper’s authors do not believe this is particularly due to lack of interest or some difficulty regarding such kind of investigation, rather to underestimating the effect of actual TL on musculoskeletal system health of this population.”

Point 6: L96-97 “players were not allowed to participate in any training plans other than the one monitored within this study”. How was this controlled for? “But God demonstrates his own love for us in this: While we were still sinners, Christ died for us.” – Romans 5:8

Response 6: Detail was provided as follows:

“… (ii) players were not allowed to participate in any training plans other than the one monitored within this study (players’ families were involved in controlling for this)…”

Point 7: L114 “To analyse the differences among the periods taking or not into account playing position, it were recruited only players using the RPE – namely the CR10 – since the previous season.”. This sentence is slightly confusing, suggest rephrasing.

What does ‘CR10’ refer to? Suggest providing a reference if this is a standard protocol (e.g., as in L174).

Response 7: Sentence was re-phrased as follows:

“Only players using the RPE scale – a 1-to-10 scale, also known as CR10 [33] – for at least one season were recruited.”

Point 8: L135 “in order for the pelvis to”.

Response 8: Sentence was re-phrased as follows:

“Their lower back was as close as possible to the stadiometer in order for the pelvis to be attached to the stadiometer and, if possible, shoulders to touch it.”

Point 9: Figure 1-3 In the reviewed version of the manuscript, the text in all three figures was too small to read. Suggest carefully considering the font size of the text to enhance readability.

Response 9: It were added higher resolution figures.

Point 10: L290-300 A major finding discussed here is that training loads were highest early- and mid-season. Following the line of argument presented that this higher load may be detrimental for musculoskeletal health, did the authors also observe a decrease in health parameters? For example, was there an increase in injury rate, decreased performance, etc.?

Response 10: Lack of measurements of health and fitness variables was acknowledged as a study limitation as follows:

“Finally, along with the higher TLs detected in early- and mid-season, specific health (e.g., injury rate and well-being indices such as Hooper Index [50]) and (further) fitness (e.g., mid- and end-season VO2max) variables could be assessed.”

Point 11: L308 What external load metrics would be most useful to measure and assess, alongside the internal load metrics presented in the present study? Suggest elaborating further to inform future work.

Response 11: We thank expert reviewer for his/her suggestion. Information was provided as follows:

“Thirdly, it is suggested to consider external load (e.g., Player LoadTM and acceleration-estimated metabolic cost [5, 49]) in this age group monitoring by utilizing some devices like wearable inertial monitoring units or electronic performance tracking systems.”

We hope that the manuscript has now reached the standard necessary for formal acceptance in Healthcare.

We look forward to hearing from you.

Best regards

Reviewer 2 Report

Nobari and colleagues have written a sound manuscript, however, the overall relevance and applicability of the findings are not well presented. I would suggest revising the narrative and ensuring the data reflects the narrative.

Abstract:

The authors have not outlined the results here, the abstract summarizes the method, then culminates in the conclusion, where are the key findings? And by meaningful- do your mean statically significant?

Ln 27- do you mean w17?

Introduction

Ln 63-  ‘Weakly’ I’m assuming this is a typographical error- Weekly?

Method

Ln 88- This data would be better presented in a table.

Ln 88- Could you specify how the participants were classified as elite? Perhaps pre-elite youth athletes may be a more appropriate term.

Ln 156- Did you only assess fitness pre-season? Assessments mid- and post-season would have strengthened your data set.

Ln 172- Did you record how many sessions per week/ the duration of these sessions etc? If so, this data should be included.

Results & Discussion

Overall:

While you have noted statistical significance across training load, this is to be expected with periodization and proper exercise programming

The direction of the paper is not clear, for example, if you’re case is for high load and incidence of injury in youth sports, did you document the occurrence of injury? Or were there any other performance indicators? I would suggest revisiting the discussion and focusing on what this data means for the youth athlete? Currently, this does not come across well.

Other perceptual measures of sleep, general well-being would have dramatically enhanced this study, something to consider for future studies.

Author Response

Response to Reviewer 2 Comments

Does the introduction provide sufficient background and include all relevant references?

(x) Must be improved

Please, read below comments to specific points.

Is the research design appropriate?

(x) Must be improved

Please, read below comments to specific points.

Are the methods adequately described?

(x) Must be improved

Please, read below comments to specific points.

Are the results clearly presented?

(x) Must be improved

Please, read below comments to specific points.

Are the conclusions supported by the results?

(x) Must be improved

Please, read below comments to specific points.

Point 1: The authors have not outlined the results here, the abstract summarizes the method, then culminates in the conclusion, where are the key findings? And by meaningful- do your mean statically significant?

Response 1: We thank expert reviewer for his/her suggestion. Key findings and significances were added.

Point 2: Ln 27- do you mean w17?

Response 2: Yes, mistake was amended.

Point 3: Ln 63- ‘Weakly’ I’m assuming this is a typographical error- Weekly?

Response 3: Yes, mistake was amended.

Point 4: Ln 88- This data would be better presented in a table.

Response 4: A new Table 1 with players’ anthropometric, maximum oxygen uptake and maturity status variables was added.

Point 5: Ln 88- Could you specify how the participants were classified as elite? Perhaps pre-elite youth athletes may be a more appropriate term.

Response 5: Players were defined “elite”, because they competed “in the Iran U14 national team competitions”.

Point 6: Ln 156- Did you only assess fitness pre-season? Assessments mid- and post-season would have strengthened your data set.

The direction of the paper is not clear, for example, if you’re case is for high load and incidence of injury in youth sports, did you document the occurrence of injury? Or were there any other performance indicators?

Other perceptual measures of sleep, general well-being would have dramatically enhanced this study, something to consider for future studies.

Response 6: Lack of measurements of health and fitness variables was acknowledged as a study limitation as follows:

“Thirdly, it is suggested to consider external load (e.g., Player LoadTM and acceleration-estimated metabolic cost [5, 49]) in this age group monitoring by utilizing some devices like wearable inertial monitoring units or electronic performance tracking systems. Finally, along with the higher TLs detected in early- and mid-season, specific health (e.g., injury rate and well-being indices such as Hooper Index [50]) and (further) fitness (e.g., mid- and end-season VO2max) variables could be assessed.”

Point 7: Ln 172- Did you record how many sessions per week/the duration of these sessions etc? If so, this data should be included.

Response 7: Matches and training sessions are already there in Table 2. Average duration of training sessions was added to Table 2.

Point 8: While you have noted statistical significance across training load, this is to be expected with periodization and proper exercise programming.

Response 8: We thank expert reviewer for his/her suggestion. Yet, we did not have access to the type of adopted training. We know only that the whole research period was during the competition season.

We hope that the manuscript has now reached the standard necessary for formal acceptance in Healthcare.

We look forward to hearing from you.

Best regards